# How Assad changed population growth in Sweden and Norway: Syrian refugees' impact on Nordic national and municipal demography

**Marianne Tønnessen**[1]*, **Siddartha Aradhya**[2]◉, **Eleonora Mussino**[2]◉

1 Norwegian Institute for Urban and Regional Research, Oslo Metropolitan University, Oslo, Norway,
2 Stockholm University Demography Unit, Stockholm, Sweden

◉ These authors contributed equally to this work.
* mariton@oslomet.no

**Data Availability Statement:** All relevant data are within the manuscript and its Supporting information files. All the data used in the main analyses (i.e. on the national level) are shown in Table 1, and the input data are available as a

## Abstract

In an increasingly interconnected world, the demographic effects of wars are not confined only to war zones and neighbouring areas; wars and conflicts may also change populations far away. Without the war in Syria under President Assad and the associated mass exodus of Syrian refugees, the population trends in distant countries like Sweden and Norway over the last few years would have been different. We create hypothetical scenarios of the population developments in Sweden and Norway *without* a war in Syria from 2011 onwards, where excess immigration due to the war and associated excess births are removed. The results indicate that population growth in 2016 would have been roughly 36% lower in Sweden and 26% lower in Norway without the Syrian war. The number of births in 2017 would have been about 3% lower in Sweden and 1% lower in Norway. One in ten municipalities would have had a population decline in 2016 instead of a population increase, and the largest immigrant group in Sweden by January 2019 would still be of Finnish origin.

## Introduction

The warfare in Syria has had dramatic and devastating consequences inside Syria as well as in several neighboring countries. Compared with this, Nordic societies have been largely unaffected by the warfare. But even in a corner of Europe far away from the Middle East, the Syrian war has put its mark on societies, mainly through the Syrian refugees who have arrived since the war broke out in 2011. This article aims at examining how conflicts can have global implications by estimating how refugee flows caused by the war in Syria under President Bashar al-Assad have affected the Swedish and Norwegian population growth and distribution.

The literature on demography of conflict has to a large degree been concerned with the effect of wars on mortality, and methods for estimating excess deaths due to wars in countries such as Cambodia, Congo, Iraq, Rwanda, Vietnam and former Yugoslavia [1–15]. Several studies have also been concerned with armed conflicts' consequences on births and fertility

Supporting information file. The same is the case for the additional analyses on demographic effects in municipalities. The only restriction on sharing these data concerns cells with few observations, in data not already published by the statistical agencies. This is only the case for the data on the number of Syrians in Swedish municipalities. Thus, in the uploaded Supporting information, numbers from 1-9 have been replaced with '<10', for one or several of the years in 10 Swedish municipalities.

**Funding:** Financial support for the research has been provided by Statistics Norway (www.ssb.no) (to MT) as well as by Oslo Metropolitan University (www.oslomet.no), the Swedish Research Council through the Swedish Initiative for Research on Microdata in the Social and Medical Sciences (https://simsam.nu/about-simsam/) (grant 340-2013-5164) and the Swedish Research Council for Health, Working Life and Welfare FORTE (https://www.government.se/government-agencies/swedish-research-council-for-health-working-life–forskningsradet-for-arbetsliv-halsa-och-valfard-forte/) (grant 2016-07105)(to all authors). The sponsors did not play any role in the study design, data collection and analysis, decision to publish, or preparation of the manuscript.

**Competing interests:** The authors have declared that no competing interests exist.

[16–26] and emigration [27–30], sometimes in order to create scenarios without deaths due to war, for comparison [2, 14, 31, 32]. Consistently, the focus has been on the demographic consequences on countries experiencing warfare.

Another body of research has focused on the general determinants of international migration [33–40]. These have mainly been concerned with determinants such as differences in economic conditions, as well as migration policies, networks of migrants in destination, geographical distance, colonial ties and common language, although some also show how war and conflict explain parts of the immigration into Western countries [41–43]. Furthermore, substantial research have aimed at showing the effects of migration on the receiving societies, also in demographical aspects (for instance for fertility [44], life expectancy [45, 46], population size and structure [47], or all these aspects [48]).

This article brings geographically distant effects into the literature on demography of conflict, as well as tools from the demography of conflict into research on conflict as a determinant of migration and migration's effects in destination. We do this by creating hypothetical scenarios of the population growth in Sweden and Norway without the Syrian war and calculate how these hypothetical trends differ from the observed developments. The no-war-scenarios take into account that there would still be some immigration of Syrians in a situation without war, and also consider that Syrian immigrants contribute to population growth through births after arrival.

Our results indicate that the war in Syria had a large effect on population growth in these two Nordic countries, particularly in 2016 when many of the refugees arriving in 2015 were granted permission to stay. Without the Syrian war, population growth at the country-level in 2016 would have been roughly 36% lower in Sweden and 26% lower in Norway. Also, the Finnish-born would still have been the largest immigrant group in Sweden by January 2019. Moreover, Syrian migration altered the population trajectories in many municipalities threatened by depopulation. Roughly one in ten municipalities in both countries that otherwise would have experienced population decline in 2016, instead saw population growth. As such, the demographic effects of the Syrian war on Norway and Sweden were not limited to urban settings but were felt throughout the countries.

## The war in Syria and its refugees

In 2011, political unrest and civil uprising erupted in Tunisia and quickly spread to several other North African and Middle Eastern countries. To date, this movement, which became known as the Arab Spring, has toppled three governments, led to at least six major governance changes, and ignited two interstate wars. Although initially these developments were met with great optimism in the international community, positive sentiments quickly faded as these conflicts escalated into global geo-political crises.

The war in Syria, in particular, is one resulting crisis that has had a profound international impact. In March 2011 President Assad ordered crackdowns and military sieges on Arab Spring protesters, which led to the Syrian civil war. Since 2011 this war has developed from a brutal internal conflict to an international proxy war between global powers. Between 1989 and 2017, the conflict was the largest in terms of fatalities in state-based conflict and second in total number of fatalities after the crisis in Rwanda [49], leading the conflict to be declared "the biggest humanitarian and refugee crisis of our time" by Filippo Grandi, the United Nations High Commissioner for Refugees [50]. Approximately six million Syrian refugees have fled the country and another six million remain internally displaced. Although neighboring countries, such as Turkey, Jordan, Lebanon and Iraq, received an overwhelming majority of the Syrian

refugees, a sizeable share made it to Europe. As of 2017, UNHCR found that 15 percent of externally displaced Syrians ended up in the European Union, with 560,000 arriving in Germany—or approximately 50 percent of those that came to the EU [51].

In Sweden and Norway, the Syrian immigrant population increased substantially between 2011 and 2018. In 2011, the Syrian-born population accounted for only 1.5 percent and less than 0.5 percent of the foreign-born populations in Sweden and Norway, respectively. In Sweden, there were only a few hundred asylum applications annually from Syria until 2012, after which the number asylum seekers increased dramatically for the following few years [52]. In 2015, the number of asylum applications from people fleeing Syria surpassed 50,000, causing the government to introduce border controls to reduce the number of asylum seekers from entering [53]. In Norway this process was somewhat delayed. Immigration from Syria remained at relatively low levels until 2015 when there was a one-year spike in the number of asylum seekers [54]. By 2018, the share of the foreign-born population who originated from Syria had increased to approximately 9 percent in Sweden and 3.5 percent in Norway.

## Methods

The demographic effect of a war for a refugee-receiving country is not identical to the net migration from the war-torn country after the outbreak of the war, for two reasons: First, there would likely have been some immigration from this country even without a war, and second, immigrants also contribute to the population growth in destination by giving births. Our approach takes both these factors into account.

In this article, the demographic effect of the Syrian war is calculated by creating no-war counterfactual scenarios where net migration of Syrians to Sweden and Norway is assumed to follow an extrapolation of the 2003–2010 trends. By subtracting these counterfactual immigration flows from the observed numbers, we identify the *excess immigrants* that arrived as a result of the war. These are additional immigrants compared to what we would expect to find if the pre-war immigration trends of Syrians were to continue, and can be defined (in line with the definition of 'excess deaths' in Daponte [12], p.53) as 'migration above and beyond what one would have expected in the population had the demographic conditions and/or trends present just prior to the war continued.' Similarly, *excess births* are estimated as the difference between all children born to Syrian-born women in Sweden or Norway and the estimated corresponding number of births in a no-war scenario. This method is further elaborated below.

### The no-war scenarios, excess net migration and excess births

The no-war scenarios are calculated for Sweden and Norway separately. First, we extrapolate the trends in net migration of Syrians to Sweden and Norway from 2003 (first available year in our Norwegian data) through 2010. Already before the war, Sweden had a considerably higher immigration of Syrians than what Norway experienced. In both countries, annual net migration from Syria increased in the period 2003–2010, from 541 to 1,159 for Sweden and from 44 to 59 in Norway.

By linear extrapolation, we create hypothetical no-war scenarios for net migration from 2011 onwards (below we discuss this assumption). In these scenarios, Sweden would have a no-war net migration of 1,865 Syrians in 2018, whereas the corresponding number for Norway would be 76.

By subtracting the net migration in these no-war scenarios ($NM_{NW,t}$) from the actual net migration ($NM_{A,t}$) of Syrian-born to Sweden and Norway from 2011, we estimate excess net

migration of Syrians ($NM_{E,t}$) due to the war, for year $t$:

$$NM_{E,t} = NM_{A,t} - NM_{NW,t} \qquad (1)$$

Using the no-war scenarios on net migration of Syrians, we also calculate no-war estimates on the number (stock) of Syrian immigrants living in Sweden and Norway ($S_{NW,t}$). On January 1$^{st}$ 2011 there were almost 21,000 Syrian-born living in Sweden and 1,500 in Norway. Adding annual no-war net migration to these figures gives us no-war estimates of the stock of Syrians in Sweden and Norway. Note that this does not take into account mortality among Syrian immigrants, a simplification which is discussed later in this section. Thus, for year $t$, the estimated no-war populations of Syrians in Sweden or Norway is calculated as the actual stock of Syrians per 01.01.2011 ($S_{A,\,2011}$) plus the cumulated no-war net migration from 2011 onwards:

$$S_{NW,t} = S_{A,2011} + \sum\nolimits_{s=2011}^{t} NM_{NW,s} \qquad (2)$$

To calculate the estimates of excess births, we start with the no-war estimates on the stock of Syrians ($S_{NW,t}$). The no-war stock estimates are divided by the actual stock of Syrians ($S_{A,t}$), to show the share of the non-war stock compared to the actual stock. This is done for every year from 2011. Next, we apply these shares to the annual number of actual births to Syrian-born women in Sweden and Norway ($B_{A,t}$), to get annual no-war estimates on the number of births to the no-war stocks of Syrians. The remaining births are considered 'excess births' due to the Syrian war. This implies that we assume that the no-war Syrian immigrants would have generated the same number of births per person as all actual Syrian immigrants did—an assumption that is also discussed below. Thus, the excess births ($B_{E,t}$) due to the Syrian war are calculated as:

$$B_{E,t} = B_{A,t} - \left( B_{A,t} * \frac{S_{NW,t}}{S_{A,t}} \right) \qquad (3)$$

The sum of the excess net immigration of Syrians and the excess births constitutes the excess population growth due to the war in Syria, $PG_{E,t}$:

$$PG_{E,t} = NM_{E,t} + B_{E,t} \qquad (4)$$

where $PG_{E,t}$ is the excess population growth in a certain year (in absolute numbers), $NM_{E,t}$ is the excess net migration the same year, and $B_{E,t}$ are the excess number of births to Syrian women. Hence, the no-war population growth ($PG_{NW,t}$) in Sweden or Norway is estimated as

$$PG_{NW,t} = PG_{A,t} - PG_{E,t} \qquad (5)$$

where $PG_{A,t}$ is the actual total population growth in these countries whereas $PG_{E,t}$ is the excess population growth due to the Syrian war.

The no-war population of Sweden and Norway in year t ($POP_{NW,t}$) is calculated as the actual population in 2011 plus the sum of the annual no-war population growth ($PG_{NW,t}$).

$$POP_{NW,t} = POP_{A,2011} + \sum\nolimits_{s=2011}^{t} PG_{NW,s} \qquad (6)$$

Further, data also allow us to calculate demographic effects of the Syrian war at the municipality level. This no-war scenario is calculated by assuming that, for each year, each municipality's share of no-war Syrians would be the same as the municipality's share of all Syrians in Sweden and Norway, respectively. Hence, for every year we find each municipality's actual share of all Syrians in the country, and apply this share to the number of excess Syrians each

year on the national level, in order to estimate non-war population growth in each municipality in the relevant year, as follows:

$$PG_{NWM,t} = PG_{AM,t} - PG_{E,t} * \frac{S_{AM,t}}{S_{A,t}} \tag{7}$$

where $PG_{NWM,t}$ is the no-war population change in the municipality (in absolute numbers), $PG_{AM,t}$ is the actual population change in the municipality, $PG_{E,t}$ is the excess population growth (at the national level) calculated in Eq (4), $S_{AM,t}$ is the actual stock of Syrian-born immigrants in the municipality and $S_{A,t}$ is the actual stock of Syrian-born immigrants nationally.

**Discussion of the assumptions.** As shown above, the method used to create no-war scenarios in this article builds on only a few, relatively simple assumptions. In the following, we discuss these assumptions, and in the discussion chapter we elaborate on the assumptions that are *not* made; for instance, about the war's effects on immigration policies, on immigration from other countries, or on the demographic behavior of natives and other immigrants in Sweden and Norway.

Perhaps the most crucial assumption in our approach, is the linear extrapolation of the 2003–2010 *net migration trends*. We prolonged the average annual net migration growth (in numbers) from 2003 to 2010, but we could have used other types of extrapolations, such as exponential, or kept the net migration constant into the future (based on the level in 2011 or some average of the years before the outbreak of the war). It could also have been possible to compare our no-war scenario of net migration with the net migration from similar countries that did not experience a war, an approach recommended and often used in works on demography of conflict [2, 10, 11]. To explore the plausibility of our assumption on linear extrapolation, we have compared our no-war scenario of Syrian net migration in Sweden and Norway with the actual net migration from ten countries neighbouring Syria, where some countries did and some did not experience an Arab spring and/or regime change (see the online S1 Appendix). The results from this exercise indicate that our no-war net migration assumptions lie within the range of net migration trends from similar countries. For Sweden, our no-war net migration scenario is in the higher range compared to the other countries, implying that our estimates on excess net migration may be too low and thus our estimated effects of the war on the Swedish population can be interpreted as a lower-bound. Moreover, we could have extended the time period for the extrapolation. The time period currently used started in 2003 because this was the first year in our data from Norway; however, for Sweden, we have data from 2000. To check whether it would make a difference to start in 2000 instead of 2003, we have re-estimated our results using extrapolations of the 2000–2010 trends for Sweden (instead of 2003–2010). The results from this exercise show very small differences compared to the analysis using 2003–2010: With extrapolations from 2000, the no-war stock of Syrian-born in Sweden per January 1st 2019 would be 32,827 (instead of 33,208), and the excess population size due to the war would be 163,962 (instead of 163,555).

For *births*, we assume that they are proportionally distributed between the no-war Syrians and those who arrived due to the war. This may not be the case, for several reasons. Births are not evenly distributed by sex and age—they are mostly confined to women age 20–39. If we compare women age 20–39's share of all the Syrian-born immigrants in Sweden and Norway in 2011 (before the war) and in 2018, the share of women age 20–39 was clearly lower in 2017, as the refugee flows were male dominated (in 2011, 22% of the Syrian-born in Sweden and 24% of the Syrian-born immigrants in Norway were women age 20–39 whereas the corresponding figure for 2018 was 16% in both Sweden and Norway). This suggests that the crude

number of births per person would be lower among the excess immigrants. On the other hand, however, previous research has shown that immigrants' fertility is often highest right after arrival [55]. Since the excess-immigrants are relatively newly arrived, their fertility may be higher than that of 'no-war' Syrians. This pulls in the other direction, indicating that our estimates of excess births may be too low, and that the actual effect of the Syrian war may be higher than we estimate.

Finally, we have not assumed *deaths* in the no-war population of Syrian-born immigrants. For long run estimates of the demographic effects of the Syrian war, this would have posed a problem. However, the Syrian-born population in Sweden and Norway is still relatively young, with very few persons in mortality-intensive ages; in both 2010 and 2017, 99% of Syrian-born individuals in Sweden and Norway were below 80 years. Estimating deaths properly would require an estimated no-war population broken down by sex and age, and for such a young population this would add much complexity to the method without really changing the results (for instance, in a no-war Syrian population of 30,000, if 1% are 80 years or more and has a mortality of 0.2 (which is relatively high, particularly since immigrants and refugees are likely to experience lower mortality than natives [56]), this corresponds to 60 deaths per year, or 0,2% of the no-war Syrian population). Since deaths would (slightly) reduce the size of the no-war population, assuming no mortality in this group also means that our estimates on the effect of the Syrian war may be slightly downward biased.

In these calculations, we have used net migration and not (gross) immigrations. Using net migration has some advantages: It is a simple way to handle emigration, since emigrations are already deducted in the net migration figures. In addition, data on net migration may be more readily available in many contexts compared with data on each separate flow. Further, net migration shows the contribution from migration to population change, which is what we need in order to estimate the growth in the no-war population. Treating immigrations and emigration separately could perhaps open up for more sophisticated models, however in the case of Syrian immigrants in Sweden and Norway the emigration rate is low; annually, less than 1% of the Syrian-born individuals emigrate.

Our analysis at the municipality level assumes that the no-war populations of Syrians would have been distributed similarly as the observed Syrian population each year. This may not be the case, at least not for Norway; when the large number of Syrians arrived in the fall of 2015 all municipalities in Norway were encouraged to accept refugees, even those with no existing Syrian-born population [57], and quite a few were rural municipalities with shrinking populations. Newly-arrived Syrians are now to a much higher degree living in rural areas than immigrants in general and also compared with the Syrians who lived in Norway before 2011 [54]. Thus, it is reasonable to suggest that the no-war population would be living in somewhat less rural areas in Norway than what our method assumes. This suggests that we have a somewhat too low estimate on the number of Norwegian municipalities where population decline was turned into population increase due to the Syrian war. In Sweden, refugees can decide where to live, however many of the new cohorts of Syrian-born were placed by the Migration Board. This is why the Syrian born population in Sweden in 2017 was more geographically distributed than in the pre-war period [53].

Despite the uncertainty associated with our assumptions, using a method with relatively few assumptions has several advantages. First, a parsimonious method is more transparent and easier to explain. Second, the simplicity of the model matches the uncertainty in this field; using an elaborate model might create a false impression of accuracy in a world where all migration flows, and in particular refugee flows, are affected by so many unpredictable factors. Even with the best specified model with respect to demographic change, there will always be

political and other stochastic factors affecting the flows that are impossible to take into account, as further elaborated in the discussion chapter.

## Similarity to other methods in conflict demography

The method outlined above has, to our knowledge, not been used in previous studies. As mentioned in the introduction, the methods in conflict demography has to a large extent focused on mortality and excess deaths in the country of war, and migration flows have usually been estimated in more discretionary ways or taken from other sources, often as part of an approach to estimate war mortality and deaths. In an overview of different approaches used to estimate civilian causalities associated with the armed conflict in Iraq, Daponte [11] lists different possible methods which also could have been applied to our setting: (a) *Tallies of deaths*, adding up the number of deaths that occur in incidents of violence as recorded by various sources, (b) *sample (household) surveys to estimate change in mortality rates from a pre-war period*, extrapolating the survey results to the general population, and (c) *demographic analyses*, i.e. making a no-war population projection from some starting point before the war, and compare with real census data (usually collected after the war). The difference is interpreted as the effect of the war.

Since we are interested in estimating excess net migration due to a war in another country, the parallels to these three approaches could be to ask either (a) all or (b) a sample of post-war immigrants from Syria whether they came due to the war, and (in b) apply this share to all post-war immigrants from Syria. For both (a) and (b), we would have to also ask about children born to these immigrants in Sweden and Norway, to be able to estimate the total effects on population size. The approach we use in this article, is a parallel to (c). However, since our model is tailored to focus on migration flows and their effects far away from the war and we want to keep the model simple, our model is more parsimonious than the standard analyses often used to estimate the demographic effects in a country affected by war. For instance, we do not run full population projections, as we do not assume changes in the general (Swedish and Norwegian) population's mortality and fertility when only the migration flows are assumed to change.

## Data

Unlike most other studies on the demography of conflict, the data availability for this study is relatively good. We use data from the Swedish and the Norwegian population register, which is considered of high quality, and which covers every person who has a documented residence in the country. Most of the data used in our analyses are publicly available at the web-based statistical databases of Statistics Sweden and Statistics Norway (the only data we needed to obtain separately, were the births to Syrian-born women (Norway and Sweden) and the number of Syrian-born persons in Swedish municipalities). Although the register data in both of these countries is high-quality, they are not entirely comparable. In Sweden, people have to (intend to) stay in the country for at least 12 months to be registered in the population register, whereas the limit in Norway is 6 months (and thus the Norwegian figures include some more short-term migrants). Also, Sweden mainly uses the category 'foreign born', including also people born abroad to Swedish-born parents, whereas Norway uses the category 'immigrants', defined as people born abroad with all parents and grandparents born abroad. However, these differences are not a major concern for our study since we conduct our analyses for Sweden and Norway separately.

Table 1 shows the input data (shaded columns) and how they are used to calculate no-war scenarios and assess the effect of the Syrian war on the population growth in Sweden (upper panel) and Norway (lower panel).

**Table 1. Calculation of no-war scenarios and excess population growth due to the Syrian war, Sweden (upper panel) and Norway (lower panel).**

| Sweden | | | | | | | | |
|---|---|---|---|---|---|---|---|---|
| A | B | C | D | E | F | G | H | I |
| *Actual net migration of Syria-born (NMA)* | *No-war net migration of Syria-born (NMNW)* | *Excess net migration (NME)* | *Actual stock of Syrian-borns (SA)* | *No-war stock of Syrian-borns (SNW)* | *Actual births to Syrian women (BA)* | *No-war births to Syrian women (BNW)* | *Excess births (BE)* | *Excess population growth (PGE)* |
| (Statistics Sweden) | Extrapolation of 2003–10 trends | NMA,t-NMNW,t | (Statistics Sweden) | For 2011: SA,2011 Later: SNW,t-1+ NMNW,t-1 | (Statistics Sweden) | BA,t * (SNW,t /SA,t) | BA,t—BNW,t | NME,t+BE,t |
| **2003** 541 | | | | | | | | |
| **2004** 509 | | | | | | | | |
| **2005** 624 | | | | | | | | |
| **2006** 1,010 | | | | | | | | |
| **2007** 504 | | | | | | | | |
| **2008** 612 | | | | | | | | |
| **2009** 897 | | | | | | | | |
| **2010** 1,159 | | | | | | | | |
| **2011** 1,653 | 1,247 | 406 | 20,758 | 20,758 | 691 | 691 | 0 | 406 |
| **2012** 5,211 | 1,336 | 3,875 | 22,357 | 22,005 | 663 | 653 | 10 | 3,886 |
| **2013** 14,263 | 1,424 | 12,839 | 27,510 | 23,341 | 914 | 775 | 139 | 12,978 |
| **2014** 25,914 | 1,512 | 24,402 | 41,748 | 24,765 | 1,327 | 787 | 540 | 24,942 |
| **2015** 30,400 | 1,600 | 28,800 | 67,671 | 26,277 | 2,002 | 777 | 1,225 | 30,024 |
| **2016** 51,290 | 1,689 | 49,601 | 98,216 | 27,877 | 3,177 | 902 | 2,275 | 51,877 |
| **2017** 22,028 | 1,777 | 20,251 | 149,418 | 29,566 | 4,289 | 849 | 3,440 | 23,691 |
| **2018** 13,850 | 1,865 | 11,985 | 172,258 | 31,343 | 4,605 | 838 | 3,767 | 15,752 |
| | | | | | | *Excess population size by 01.01.2019:* | | *163,555* |

| Norway | | | | | | | | |
|---|---|---|---|---|---|---|---|---|
| A | B | C | D | E | F | G | H | I |
| *Actual net migration of Syrian citizens[1] (NMA)* | *No-war net migration of Syrian citizens (NMNW)* | *Excess net migration (NME)* | *Actual stock of Syrian immigrants, 1. January[1] (SA)* | *No-war stock of Syrian immigrants, 1. January (SNW)* | *Actual births to Syrian women (BA)* | *No-war births to Syrian women (BNW)* | *Excess births (BE)* | *Excess population growth (PGE)* |
| (Statistics Norway) | Extrapolation of 2003–10 trends | NMA,t-NMNM,t | (Statistics Norway) | For 2011: SA,2011 Later: SNW,t-1+ NMNW,t-1 | (Statistics Norway) | BA,t * (SNW,t /SA,t) | BA,t—BNW,t | NME,t+BE,t |
| **2003** 44 | | | | | | | | |
| **2004** 47 | | | | | | | | |
| **2005** 61 | | | | | | | | |
| **2006** 54 | | | | | | | | |
| **2007** 57 | | | | | | | | |
| **2008** 81 | | | | | | | | |
| **2009** 48 | | | | | | | | |
| **2010** 59 | | | | | | | | |
| **2011** 56 | 61 | -5 | 1,497 | 1,497 | 69 | 69 | 0 | -5 |
| **2012** 360 | 63 | 297 | 1,571 | 1,558 | 65 | 64 | 1 | 297 |
| **2013** 830 | 65 | 765 | 2,009 | 1,621 | 82 | 66 | 16 | 780 |
| **2014** 2,118 | 68 | 2,050 | 3,080 | 1,687 | 107 | 59 | 48 | 2,099 |
| **2015** 3,986 | 70 | 3,916 | 5,450 | 1,754 | 177 | 57 | 120 | 4,036 |
| **2016** 11,184 | 72 | 11,112 | 9,710 | 1,824 | 350 | 66 | 284 | 11,396 |

*(Continued)*

**Table 1.** (Continued)

| | | | | | | | | |
|---|---|---|---|---|---|---|---|---|
| **2017** | 6,874 | 74 | 6,800 | 20,823 | 1,896 | 689 | 63 | 626 | 7,426 |
| **2018** | 3,744 | 76 | 3,668 | 27,392 | 1,970 | 714 | 51 | 663 | 4,331 |
| | | | | | | *Excess population size by 01.01.2019:* | | *30,361* |

Sources for actual numbers: Statistics Sweden and Statistics Norway

[1]Due to data availability, figures for Norway are by citizenships for actual net migration, and by country of birth for actual stock of Syrians.

## Results

The war in Syria affected the population in Sweden and Norway in several ways. Here, we focus on the population growth and size, the number of births, the ranking of immigrant groups, and population growth at the municipality level.

First, the *population growth* increased substantially due to the Syrian war. According to our estimates (Table 1, right column) the excess population growth was particularly high in 2016, when many of the refugees arriving in the fall of 2015 got their permission to stay and thus were recorded as immigrated. Without the Syrian war, population growth in Sweden would have been around 29% lower in 2015, 36% lower in 2016 and 19% lower in 2017. In Norway, the growth would have been approximately 26% lower in 2016 and 20% lower in 2017. The effects are depicted in Fig 1.

Sweden's population would have increased between 2011 and 2017 without the mass immigration of Syrian refugees, but to a significantly smaller extent. Specifically, in 2016 the population growth in Sweden stood at more than 144,000 –which is an historical record in annual population growth. Without the Syria war, our estimates suggest that the Swedish population growth in 2016 would have been about 50,000 lower. In Norway, the growth rate of the population declined after 2011–2012, which was a period with relatively high fertility, low mortality, and high immigration due to a favorable economic cycle. The decrease in the Norwegian population growth—mainly due to lower labour migration and falling birth rates—was slowed because of the Syrian war, but not enough to stop the growth from declining.

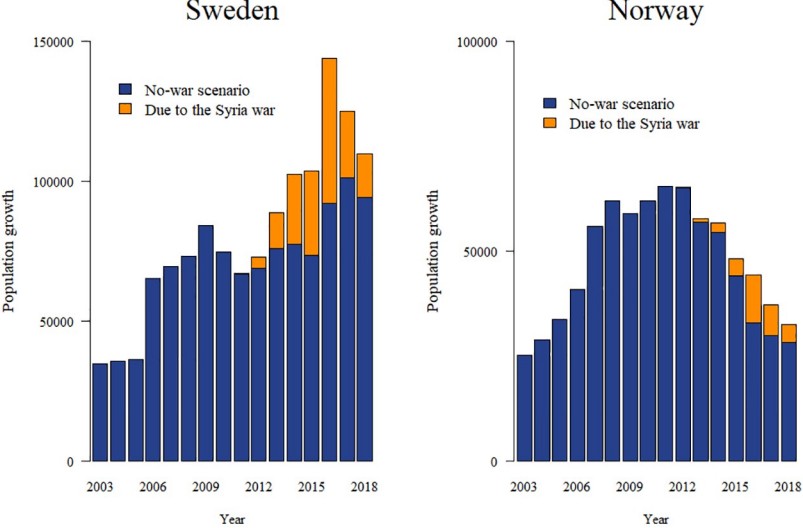

**Fig 1. Population growth in Sweden and Norway 2010–2018, estimated with and without the Syrian war.**

**Table 2. Largest groups of foreign-born/immigrants in Sweden and Norway per 1.1.2019.**

| | Sweden 2018 | | Sweden 2019 | | Norway 2018 | | Norway 2019 | |
|---|---|---|---|---|---|---|---|---|
| 1 | Syria (without the war) | 172,258 (31,343) | Syria (without the war) | 185,991 (33,208) | Poland | 98,212 | Poland | 98,691 |
| 2 | Finland | 150,877 | Finland | 147,883 | Lithuania | 38,371 | Lithuania | 39,300 |
| 3 | Iraq | 140,830 | Iraq | 144,035 | Sweden | 35,813 | Sweden | 35,586 |
| 4 | Poland | 91,180 | Poland | 92,759 | Somalia | 28,754 | Syria (without the war) | 30,795 (2,046) |
| 5 | Iran | 74,096 | Iran | 77,386 | Syria (without the war) | 27,392 (1,970) | Somalia | 28,642 |
| 6 | Somalia | 66,369 | Somalia | 68,678 | Germany | 24,445 | Germany | 24,567 |
| 7 | Yugoslavia (rest) | 65,877 | Yugoslavia (rest) | 65,124 | Iraq | 23,118 | Iraq | 23,228 |
| 8 | Bosnia and Herzegovina | 58,880 | Bosnia and Herzegovina | 59,395 | Eritrea | 21,747 | Eritrea | 22,560 |
| 9 | Germany | 50,863 | Afghanistan | 51,979 | Philippines | 21,383 | Philippines | 22,272 |
| 10 | Turkey | 48,299 | Germany | 51,140 | Pakistan | 20,372 | Pakistan | 20,674 |
| 11 | Afghanistan | 43,991 | Turkey | 49,948 | Thailand | 19,507 | Thailand | 20,383 |
| 12 | Norway | 42,028 | Thailand | 42,394 | Denmark | 19,267 | Denmark | 19,150 |
| 13 | Thailand | 41,240 | Eritrea | 42,300 | Iran | 17,728 | Iran | 18,075 |
| 14 | Denmark | 40,563 | Norway | 41,747 | Russia | 17,480 | Russia | 17,783 |
| 15 | Eritrea | 39,081 | Denmark | 40,011 | Afghanistan | 16,782 | Afghanistan | 17,023 |
| 16 | China | 31,333 | India | 35,234 | United Kingdom | 14,261 | Romania | 14,505 |
| 17 | India | 29,673 | China | 33,288 | Romania | 14,206 | United Kingdom | 14,485 |
| 18 | Romania | 29,546 | Romania | 31,040 | Vietnam | 13,973 | Vietnam | 14,100 |

The numbers in normal fonts show the actual numbers, the numbers in parenthesis and italics for Syrians show the no-war scenario. Sources for actual numbers: Statistics Sweden and Statistics Norway

Consequently, the *population size* in Sweden and Norway would also have been lower without the war in Syria. According to our estimates, the Swedish population on the 1st of January 2018 would have been nearly 150,000 lower, or 9.97 million inhabitants instead of the actual 10.12 million. This implies that without the Syrian war, the Swedish society would not have passed the threshold of 10 million inhabitants like they did in January 2017 [58]. Instead, the Swedish population would have surpassed 10 million some time in 2018, and according to our estimates the Swedish population on the 1st of January 2019 would stand at almost 10,070,000 instead of 10,230,000 –a difference of 160,000. In Norway, where the population is about half of the Swedish population, the number of inhabitants on January 2019 would have been 5,298,000 without the Syrian war—about 30,000 lower than the actual population figure of 5,328,000. This is less than a fifth of the corresponding number for Sweden, showing that the Syrian war put a clearly more pronounced mark on the Swedish population compared with the Norwegian, even in relative terms.

The number of *births* in Sweden would also have been lower. According to our estimates, there would have been 3,800 fewer births in 2018, or 3% of all births in Sweden that year. In Norway the number of births in 2018 would have been roughly 650 lower (just over 1% of all births).

The Syrian war clearly also affected the composition of immigrants in Sweden and Norway (Table 2). By January 2019, the Syrian-born population became the largest immigrant group in Sweden, even surpassing the Finnish population (150,877 individuals) that had been the largest foreign-born population in the country for decades. Without the war in Syria, however, the

Syrian-born population in Sweden would have been slightly above 31,000 individuals making them the 16[th] largest group, more in line with the number of immigrants from China or India.

In Norway, there were slightly more than 30,000 Syrian-born persons in 2019, which made them the 4[th] largest immigrant group, having passed the number of immigrants from Somalia during 2018. Without the Syrian war the number of immigrants from Syria in Norway is estimated at around 2,000 in 2019, making them a smaller group than immigrants from 50 other origin countries.

Perhaps contrary to popular perception that immigration mainly affects urban populations, the demographic impact of the Syrian war was seen even in the furthest reaches of each country. According to our estimates, 42 of Norway's 428 (nearly 1 in 10) municipalities would have experienced population decline in 2016 instead of a population growth, and most of these municipalities are rural or semi-rural. Even in 2017, when the net immigration of Syrians had slightly declined, according to our estimates 24 of the municipalities with population growth would have had a decline without the Syrian war. In Sweden, the regional effects were relatively similar; 34 and 32 of Sweden's 290 municipalities would have experienced population decline instead of an increase in 2016 and 2017, respectively, in the absence of the Syrian war. This also corresponds to around 1 in 10 municipalities.

It may be surprising that the share of municipalities experiencing an increase instead of a decrease is so similar between Sweden and Norway, even though Sweden received a substantially higher number of refugees from Syria during this period. One main reason for this may be the Norwegian settlement policy for immigrants, where refugees are assigned to municipalities by a governmental agency [59], and the urge to all municipalities to accept refugees in the fall of 2015. In 2016, many Syrians were settled in Norwegian municipalities with small and in some cases declining populations.

As a robustness check, we also calculated an alternative no-war scenario where we simply used the municipality's actual population change as a starting point, and subtracted the annual increase in number of Syrian immigrants living in the municipality. This scenario does not take into account births or the fact that there would be some immigration also in the absence of a war. In this approach, the Syrian war reversed population trends in 2016 in even more municipalities; population increase instead of decrease was found in 49 (one in six) of Sweden's and 59 (one in seven) of Norway's municipalities.

## Discussion

Despite the world seemingly becoming more peaceful with a decreasing number of fatalities in organized violence since 2014, the number of active conflicts is at a historically high level [60]. Some conflicts may have large effects also outside the area where the fighting takes place. Our analyses indicate that the war in Syria did have a substantial impact on population change even far away in Sweden and Norway. Compared with all the suffering and the upheaval of societies both inside Syria and in the neighbouring countries, the effects in Sweden and Norway are minor. But still these results show that conflict and wars can have noticeable impacts even in very distant parts of the world.

The estimated effects in Sweden and Norway may be in the lowest range. As previously mentioned, most of our assumption may give estimates of the excess population that are too low rather than too high, indicating that the war in Syria may actually have affected the population in Sweden and Norway to a greater extent than our study suggests. However, there are important factors that we have not included in our estimations, mainly because of the difficulties creating credible counterfactuals to the actual situations. For instance, we do not know how the Syrian war affected immigration to Sweden and Norway from other immigrant

groups. During the period of relatively large immigration from Syria to Europe in 2015, also people fleeing from other countries came in relatively large numbers, and the political tightening of EU borders—which possibly would not have taken place without the Syrian war—also affected people fleeing from other countries, who might otherwise have come to Sweden and Norway to a larger extent. Other research has found that increased international migration flows from one area may hamper flows from other areas [61]. To investigate this, we have checked whether migration from other parts of the world to Sweden and Norway changed after the Syrian refugee influx, see the online S2 Appendix. However, we were not able to draw clear conclusions, primarily because the patterns in Sweden and Norway were markedly different.

We also do not know whether additional Middle Eastern countries would have experienced an Arab spring if the war in Syria had not broken out, with other accompanying refugee flows, and we do not know to what extent Syrian refugees filled up the UNHCR-quotas that would otherwise be filled by refugees from other countries.

Moreover, our estimations have not taken into account possible effects of the Syrian war on the demographic behavior of native Swedes/Norwegians, or among other immigrants already living in Sweden or Norway. For instance, the fertility of natives and other immigrants could be affected in several ways: The general fertility could fall if people are less tempted to give birth during a period with historically high levels of geopolitical conflict, and the increased influx of Syrian immigrants could contribute to less political concern about future population size (and hence less need to change family policies in order to increase fertility). On the other hand, some natives' fear of becoming a minority may lead to a stronger urge to have children of their own. However, Swedish and Norwegian fertility policies were relatively stable in this period, and no clear change in the general fertility trends in neither Sweden nor Norway is observed around 2015/2016 [62, 63]. For mortality, tragic news about wars might be detrimental for some people's health and possibly mortality, but on the other hand inflow of future labor and possible future care workers can be beneficial for health care in areas who have had trouble attracting enough personnel. However, we do not see changes in the general mortality and life expectancy trends in this period [64, 65].

It is also possible that the high influx of Syrians could have affected emigration. One mechanism could be found at the labor market: Reception of refugees may imply job opportunities for locals, among others at asylum and refugee centers. On the other hand, Syrian refugees have gradually entered local labor markets, where they may compete with other immigrants for the same jobs. Changed opportunities at the job market may, in turn, affect the out-migration of natives and immigrants. Another mechanism could be that the large influx of Syrians affected natives' attitudes towards immigrants in general, which may influence other immigrants' desires to stay or emigrate. Further, the warfare in Syria may negatively affect other Middle Eastern immigrants' plans of moving back to their origin. However, the emigration trends for native citizens did not change much around 2015/2016. For non-native citizens, the number of emigrations decreased after 2015 /2016 [66, 67]. If some of this latter decline is connected to the war in Syria, for instance through mechanisms suggested above, our estimates of the war's effect on population growth are, again, too low.

To sum up the above discussion, our approach does not take into account all possible effects that the war might have had on the demography of Sweden and Norway. However, we argue that many of these are either very uncertain or the (probably minor) effects pull in the same directions as our results suggest. Therefore, instead of aiming to capture the full range of possible effects, we have chosen a rather simple method, in line with the huge uncertainty described above. We could have used a more complex model, for example a full cohort component model broken down by sex and age, where in- and out-migration was

treated separately (allowing for the use of emigration rates by sex and age). With figures also broken down by duration of stay we could incorporate falling fertility by years since migration. However, it is unclear whether more detailed method would have made the scenarios more or less accurate. In a world with much uncertainty, simple scenarios may be better because they may be more easily understood, and also more easily calculated in settings with limited data.

Our estimates only show the effects in a relatively short time span after the war broke out. The long-term effects of the Syrian war on the population in Sweden and Norway are harder to estimate, because the war is not yet over (as of November 2020), and we do not yet know what emigration back to Syria will look like after the war. Calculations from Statistics Norway [68], however, suggest that for non-Western immigrants in Norway, the population size effects of fertility almost offset the contributions of mortality and emigration. This implies that the long-term effect of, say, 10,000 additional arrivals in this group, is a population which is almost 10,000 higher through the next 50 years. As we estimated above, by 2019 the Syrian war had enlarged the Swedish population by more than 160,000 and the Norwegian population by 30,000. If Syrian immigrants in Norway and Sweden were to display similar demographic behavior as other non-Western immigrants in Norway (i.e. somewhat higher fertility and higher emigration rates than the natives), we may assume that the long-term population size effect of the Syrian war (so far) will remain about 160,000 in Sweden and 30,000 in Norway in the fifty years to come.

Besides the general impacts on the population at the national level, it can be argued that the Syrian war has played an important role in regional developments in Sweden and Norway. In both countries, rural and semi-rural municipalities have been facing pressures as a result of gradual depopulation. Young people have increasingly migrated to urban centers leading to population aging concerns in these settings [69, 70]. Our study shows that at least some of these municipalities reversed this trend as a result of the Syrian immigration, and this has the potential to revitalize the local economies of some regions as long as the immigrants stay. As such, the role that the Syrian immigrants can play to help municipalities struggling with depopulation to become self-sustainable must not be understated.

## Summary and conclusion

The recent war in Syria has had numerous consequences, most notably in Syria. But even far away from the war zones, in the North Western corner of Europe, it has affected the population size and growth of Sweden and Norway, nationally as well as on the municipal level. Our estimates indicate that the Swedish population would have grown substantially slower without the excess immigration from Syria. This is particularly the case for 2016 when the actual Swedish population growth was record high; in a scenario without the war it would have been 36% lower. Similarly, in Norway the population growth in 2016 would have been 26% lower without the Syrian war, according to our estimates. In the absence of a war, the Swedish society could not have marked the passing of 10 million inhabitants like they did in 2017, Finnish-born persons would still constitute the largest immigrant group in Sweden by 1.1.2019, and the number of births in 2018 would have been 3,750 (3%) lower in Sweden and 650 (1%) lower in Norway. Around 1 in 10 of the municipalities in both Sweden and Norway would have seen a population decline in 2016 instead of a population increase.

These results are obtained using a simple but innovative method which takes into account that some migration would have taken place from Syria to Sweden and Norway even in the absence of a war, and it also accounts for births among those immigrant women estimated to have come due to the war. We do this using rich register data which is rare in conflict

demography and by developing new methods—for the national as well as the municipal level —that to our knowledge has not been published before. As such, this article brings distant effects into the literature on demography of conflict, which until now mainly has been concerned with estimating demographic effects within areas of conflict and war, and it also brings tools from demography of conflict into the literature on determinants of migration and on migration's effect in destination countries. The results illustrate that in today's world, the consequences of wars and conflicts do not necessarily stop at borders, but can also be found even in countries and municipalities far, far away.

## Supporting information

**S1 Appendix. Comparisons with net migration trends from similar countries.**
(DOCX)

**S2 Appendix. Net migration from other countries before and after 2015.**
(DOCX)

**S1 Data. Input data on national level.**
(XLSX)

**S2 Data. Input data on municipal level.**
(XLSX)

**S1 Fig.**
(TIF)

**S2 Fig.**
(TIF)

## Acknowledgments

We are grateful for valuable feedback and help from participants at Nordic Demographic Symposium 2019 as well as from Terje Skjerpen, Statistics Norway and Statistics Sweden.

## Author Contributions

**Conceptualization:** Marianne Tønnessen.

**Data curation:** Marianne Tønnessen, Siddartha Aradhya, Eleonora Mussino.

**Formal analysis:** Marianne Tønnessen.

**Methodology:** Marianne Tønnessen.

**Project administration:** Marianne Tønnessen.

**Resources:** Marianne Tønnessen, Siddartha Aradhya, Eleonora Mussino.

**Supervision:** Siddartha Aradhya, Eleonora Mussino.

**Validation:** Marianne Tønnessen, Siddartha Aradhya, Eleonora Mussino.

**Visualization:** Marianne Tønnessen.

**Writing – original draft:** Marianne Tønnessen, Siddartha Aradhya, Eleonora Mussino.

**Writing – review & editing:** Marianne Tønnessen, Siddartha Aradhya, Eleonora Mussino.

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
