## [Decision Letter · Decision Letter 0]

28 Sep 2020

PONE-D-20-23314

How the war in Syria changed the demography of Sweden and Norway

PLOS ONE

Dear Dr. Tønnessen,

Thank you for submitting your manuscript to PLOS ONE. After careful consideration, we feel that it has merit but does not fully meet PLOS ONE’s publication criteria as it currently stands. Therefore, we invite you to submit a revised version of the manuscript that addresses the points raised during the review process.

We have now received comments from two expert reviewers. The first reviewer is quite positive about your paper, while the second one is more negative. Given the importance of the topic, I would like to give you an opportunity to submit a revised version of your paper. Please try to address the reviewers' comments as fully as possible.

We look forward to receiving your revised manuscript.

Kind regards,

Semih Tumen, PhD

Academic Editor

PLOS ONE

Journal Requirements:

2.We note that you have indicated that data from this study are available upon request. PLOS only allows data to be available upon request if there are legal or ethical restrictions on sharing data publicly. For more information on unacceptable data access restrictions, please see http://journals.plos.org/plosone/s/data-availability#loc-unacceptable-data-access-restrictions.

Reviewers' comments:

Reviewer's Responses to Questions

**Comments to the Author**

1. Is the manuscript technically sound, and do the data support the conclusions?

Reviewer #1: Yes

Reviewer #2: Partly

2. Has the statistical analysis been performed appropriately and rigorously? 

Reviewer #1: Yes

Reviewer #2: No

3. Have the authors made all data underlying the findings in their manuscript fully available?

Reviewer #1: Yes

Reviewer #2: Yes

4. Is the manuscript presented in an intelligible fashion and written in standard English?

Reviewer #1: Yes

Reviewer #2: Yes

5. Review Comments to the Author

Reviewer #1: The authors explore the demographic effect of the Syrian war on Sweden and Norway by creating no-war counterfactual scenarios where net migration of Syrians to Sweden and Norway is assumed to follow an extrapolation of the 2003-2010 trend. They find that the population growth would have been roughly 36 % lower in Sweden and 26 % lower in Norway without the Syrian war particularly in 2016 when many of the refugees arriving in 2015 were granted permission to stay. They also show that approximately one in ten municipalities would have had a population decline in Sweden and Norway after 2015.

The findings in the manuscript are presented in an appropriate fashion and are supported by the data. They are also described in sufficient detail. However, it will be good to see a discussion on how the additional immigrants from Syria affect fertility and family policies, immigrant policies and local perceptions towards immigrant groups in Sweden and Norway.

Other comments:

• The authors need to discuss why they follow an extrapolation of the 2003-2010 trend instead of such as 2000-2010 trend?

• Figure 1 should be extended to be started from 2003 or earlier so that we can see that the previous trend supports no-war scenario.

• “The numbers in normal fonts show the actual numbers, whereas the numbers in parenthesis and italics for Syrians show the no-war scenario” in title of Table 2 should be removed from the title to the source.

Reviewer #2: This paper aims to estimate the hypothetical population numbers had the Syrian refugee influx not occurred in Sweden and Norway. To do so, the authors first calculate the increase in the number of Syrian migrants as a result of the influx by extrapolating the pre-conflict period (2003-2010) Syrian migrant trends to the period 2011-2017. Then, they calculate the excess births as a result of the increase in the number of Syrian migrants considering that the Syrian females’ birth rate is traditionally higher than the native females’ birth rate. Finally, they subtract these excess numbers from the actual population to obtain hypothetical population numbers for each country. The paper is well written and structured overall, however, there are fundamental issues related to the research question and the methodology. I list my concerns and suggestions below:

1- The title of the paper does not directly reflect the core point of the paper, which is the change in the population numbers due to Syrian migrant influx into Sweden and Norway.

2- What paper does is a simple extrapolation analyses using the pre-treatment Syrian migration trends in the respective countries and estimate counterfactual population numbers using these trends in the case of no Syrian conflict. I believe that this is much of a mechanical result rather than a causal relationship between the Syrian refugee influx and the demographics of natives such as native birth rates, job market results, impact on existing migrants etc. Therefore, I find the contribution very weak especially for international readers.

3- What is meant by “linear extrapolation” and how it is performed is not clearly described in the paper. Are these numbers extrapolated using a linear regression model with time trend or the average annual percentage growth rate during the pre-treatment period?

4- Authors should account for a possible crowding out effect on the migrants of other nationalities. This can be performed by employing a similar extrapolation analyses for the total number of migrants from other countries considering the trends before the Syrian refugee influx.

5- Minor: Figure 1 in page 14 is missing.

6. PLOS authors have the option to publish the peer review history of their article (what does this mean?). If published, this will include your full peer review and any attached files.

Reviewer #1: **Yes: **Ahmet Ozturk

Reviewer #2: No

---

## [Author Response · Author response to Decision Letter 0]

23 Nov 2020

Reviewer #1: 

The authors explore the demographic effect of the Syrian war on Sweden and Norway by creating no-war counterfactual scenarios where net migration of Syrians to Sweden and Norway is assumed to follow an extrapolation of the 2003-2010 trend. They find that the population growth would have been roughly 36 % lower in Sweden and 26 % lower in Norway without the Syrian war particularly in 2016 when many of the refugees arriving in 2015 were granted permission to stay. They also show that approximately one in ten municipalities would have had a population decline in Sweden and Norway after 2015.

The findings in the manuscript are presented in an appropriate fashion and are supported by the data. They are also described in sufficient detail. However, it will be good to see a discussion on how the additional immigrants from Syria affect fertility and family policies, immigrant policies and local perceptions towards immigrant groups in Sweden and Norway.

Thank you, and the comment in your last sentence is an interesting point that we now elaborate more on. Not many studies are conducted on these issues, but in the revised version of the Discussion chapter, we discuss possible mechanisms for how the influx of Syrians may have affected natives’ (and other immigrants’) demographic behavior, including fertility and family policies, as well as the perceptions towards immigrants, and also how the tightening of immigrant policies may have affected migration of other immigrant groups. For the latter question we have also conducted some extra analyses, see the Supplemental File S2_Appendix. 

Other comments:

• The authors need to discuss why they follow an extrapolation of the 2003-2010 trend instead of such as 2000-2010 trend?

This is simply due to data availability, 2003 was the first year in our data set for Norway (and also the first year available for replication, since it is taken from the publicly available Statistics Norway’s web-based stat-bank). Now we make it more clear in the Methods chapter that this is the first available year for our analyses. For Sweden, we however have data from 2000. To check whether it would make a difference to start in 2000 instead of 2003, we have re-estimated our results for Sweden using extrapolations of the 2000-2010 trends (instead of 2003-2010). The results from this exercise show very small differences compared to the initial analysis. Specifically, if we use data from 2000, the no-war stock of Syrian-born per January 1st 2019 would be 32,827 (instead of 33,208), and the excess population growth due to the war would be 163,962 (instead of 163,555). Now we have added this information to the manuscript, see the ‘Discussion of the assumptions’ section in the Methods chapter.

• Figure 1 should be extended to be started from 2003 or earlier so that we can see that the previous trend supports no-war scenario.

Now this is done, see the Results chapter.

• “The numbers in normal fonts show the actual numbers, whereas the numbers in parenthesis and italics for Syrians show the no-war scenario” in title of Table 2 should be removed from the title to the source.

Now this text is moved to under the table, along with the source (see the Results chapter).

Reviewer #2: 

This paper aims to estimate the hypothetical population numbers had the Syrian refugee influx not occurred in Sweden and Norway. To do so, the authors first calculate the increase in the number of Syrian migrants as a result of the influx by extrapolating the pre-conflict period (2003-2010) Syrian migrant trends to the period 2011-2017. Then, they calculate the excess births as a result of the increase in the number of Syrian migrants considering that the Syrian females’ birth rate is traditionally higher than the native females’ birth rate. Finally, they subtract these excess numbers from the actual population to obtain hypothetical population numbers for each country. The paper is well written and structured overall, however, there are fundamental issues related to the research question and the methodology. I list my concerns and suggestions below:

1- The title of the paper does not directly reflect the core point of the paper, which is the change in the population numbers due to Syrian migrant influx into Sweden and Norway.

You are right that this is the core point of our paper (although in this revised version we discuss in more detail how natives’ and other migrants’ demographic behavior may have been affected by the Syria war and the following refugee influx). We now suggest a different title, including a subtitle: 

How Assad changed population growth in Sweden and Norway 

Syrian refugees’ impact on Nordic national and municipal demography

2- What paper does is a simple extrapolation analyses using the pre-treatment Syrian migration trends in the respective countries and estimate counterfactual population numbers using these trends in the case of no Syrian conflict. I believe that this is much of a mechanical result rather than a causal relationship between the Syrian refugee influx and the demographics of natives such as native birth rates, job market results, impact on existing migrants etc. Therefore, I find the contribution very weak especially for international readers.

We agree that the main purpose of this paper is not to establish not any causal relationship between the influx of Syrian refugees and the demographic behavior (or other behavior) of people already living in Sweden and Norway. In this version of the manuscript, we aim at making our core purpose clearer (for instance with the new title, and also in the conclusion). In addition, we have broadened the discussion of how the influx of Syrians may have affected the demography in Sweden and Norway in other ways - or in addition to - the effects we estimate, where we also discuss possible effects on for instance birth rates, attitudes toward immigrants, job market effects, and impacts on existing and other potential migrants (see the Discussion chapter, and also our response to your comment 4 below).

With this done, we firmly believe that the paper contributes to the international literature in several ways. We have not seen any other attempts to quantify effects of war on populations far away from the conflict zone. We do this, using rich register data which is rare in conflict demography, and developing new methods – for the national as well as the municipal level – that to our knowledge has not been published before. Hence, the paper contributes to bringing together the literature on demography of conflict and the literature on migration’s effect in destination countries. In addition, some of our quantified result may be interesting, also to international readers.

3- What is meant by “linear extrapolation” and how it is performed is not clearly described in the paper. Are these numbers extrapolated using a linear regression model with time trend or the average annual percentage growth rate during the pre-treatment period?

In our linear extrapolation we prolonged the average annual net migration growth (in numbers) from 2003 to 2010. Now this is explained in the Methods chapter, where we also discuss this assumption (see the subsection ‘Discussion of the assumptions’).

4- Authors should account for a possible crowding out effect on the migrants of other nationalities. This can be performed by employing a similar extrapolation analyses for the total number of migrants from other countries considering the trends before the Syrian refugee influx.

Intrigued by your comment we have now conducted analyses inspired by your suggestion: First, we examined data on the trends for the total number of migrants from other countries, and found that the trends in Sweden and Norway were remarkably different. To further examine this, we investigated the effect of two possible ‘crowding out-channels’ – the employment channel (Syrians taking jobs that would otherwise be available for other migrants) and the policy channel (large influx of Syrians leading to tightening of immigrant policies in general), since we hypothesized that these channels would affect different groups of immigrants and with a different time lag from 2015/16. However, even in this last analysis the results from Sweden and Norway differed considerably, and we did not find many clear indications for crowding out effects. We still find your comment and this exercise very relevant, so now we have included these analyses in a separate Supporting-Information-file (S2_Appendix) and refer to them in the main manuscript (see the Discussion chapter).

5- Minor: Figure 1 in page 14 is missing. 

Now Figure 1 is shown in the manuscript, see the Results chapter.

Thank you both for your useful suggestions and for taking the time to improve our paper!

---

## [Decision Letter · Decision Letter 1]

15 Dec 2020

How Assad changed population growth in Sweden and Norway Syrian refugees’ impact on Nordic national and municipal demography

PONE-D-20-23314R1

Dear Dr. Tønnessen,

We’re pleased to inform you that your manuscript has been judged scientifically suitable for publication and will be formally accepted for publication once it meets all outstanding technical requirements.

Kind regards,

Semih Tumen, PhD

Academic Editor

PLOS ONE

Additional Editor Comments (optional):

Reviewers' comments:

Reviewer's Responses to Questions

**Comments to the Author**

1. If the authors have adequately addressed your comments raised in a previous round of review and you feel that this manuscript is now acceptable for publication, you may indicate that here to bypass the “Comments to the Author” section, enter your conflict of interest statement in the “Confidential to Editor” section, and submit your "Accept" recommendation.

Reviewer #1: All comments have been addressed

Reviewer #2: All comments have been addressed

2. Is the manuscript technically sound, and do the data support the conclusions?

Reviewer #1: Yes

Reviewer #2: Yes

3. Has the statistical analysis been performed appropriately and rigorously? 

Reviewer #1: Yes

Reviewer #2: Yes

4. Have the authors made all data underlying the findings in their manuscript fully available?

Reviewer #1: Yes

Reviewer #2: Yes

5. Is the manuscript presented in an intelligible fashion and written in standard English?

Reviewer #1: Yes

Reviewer #2: Yes

6. Review Comments to the Author

Reviewer #1: (No Response)

Reviewer #2: (No Response)

7. PLOS authors have the option to publish the peer review history of their article (what does this mean?). If published, this will include your full peer review and any attached files.

Reviewer #1: **Yes: **Ahmet Ozturk

Reviewer #2: No

---

## [Editor Report · Acceptance letter]

28 Dec 2020

PONE-D-20-23314R1 

How Assad changed population growth in Sweden and Norway
*Syrian refugees’ impact on Nordic national and municipal demography*

Dear Dr. Tønnessen:

I'm pleased to inform you that your manuscript has been deemed suitable for publication in PLOS ONE. Congratulations! Your manuscript is now with our production department. 

Kind regards, 

on behalf of

Professor Semih Tumen 

Academic Editor

PLOS ONE